# Integrative Analysis of Transcriptome and Metabolome Reveals Molecular Responses in *Eriocheir sinensis* with Hepatopancreatic Necrosis Disease

**DOI:** 10.3390/biology11091267

**Published:** 2022-08-26

**Authors:** Ming Zhan, Lujie Wen, Mengru Zhu, Jie Gong, Changjun Xi, Haibo Wen, Gangchun Xu, Huaishun Shen

**Affiliations:** 1Wuxi Fisheries College, Nanjing Agricultural University, Nanjing 210095, China; 2Key Laboratory of Integrated Rice-Fish Farming Ecology, Ministry of Agriculture and Rural Affairs, Freshwater Fisheries Research Center, Chinese Academy of Fishery Sciences, Wuxi 214081, China

**Keywords:** *Eriocheir sinensis*, hepatopancreas necrosis disease (HPND), transcriptomic, metabolomic, molecular responses

## Abstract

**Simple Summary:**

There are still many doubts about the etiology and pathogenic mechanisms of Chinese mitten crab hepatopancreatic necrosis disease (HPND). In the present study, new insights into the molecular responses of diseased crabs were found by integrating transcriptomics and metabolomics. The results show that the immune system, metabolic capacity, antioxidant system, detoxification capacity, and nervous system of HPND-affected crabs were impacted, and these changes may be related to the pathological mechanism of HPND. Furthermore, we suspect that functional hypoxia may be involved in the progression of HPND, which provides new insights into the pathogenic mechanism of HPND. In conclusion, based on our findings, we further confirmed that HPND could be mainly caused by abiotic factors.

**Abstract:**

Hepatopancreatic necrosis disease (HPND) is a highly lethal disease that first emerged in 2015 in Jiangsu Province, China. So far, most researchers believe that this disease is caused by abiotic factors. However, its true pathogenic mechanism remains unknown. In this study, the effects of HPND on the metabolism and other biological indicators of the Chinese mitten crab (*Eriocheir sinensis*) were evaluated by integrating transcriptomics and metabolomics. Our findings demonstrate that the innate immunity, antioxidant activity, detoxification ability, and nervous system of the diseased crabs were affected. Additionally, metabolic pathways such as lipid metabolism, nucleotide metabolism, and protein metabolism were dysregulated, and energy production was slightly increased. Moreover, the IL-17 signaling pathway was activated and high levels of autophagy and apoptosis occurred in diseased crabs, which may be related to hepatopancreas damage. The abnormal mitochondrial function and possible anaerobic metabolism observed in our study suggested that functional hypoxia may be involved in HPND progression. Furthermore, the activities of carboxylesterase and acetylcholinesterase were significantly inhibited, indicating that the diseased crabs were likely stressed by pesticides such as pyrethroids. Collectively, our findings provide new insights into the molecular mechanisms altered in diseased crabs, as well as the etiology and pathogenic mechanisms of HPND.

## 1. Introduction

The freshwater Chinese mitten crab (*Eriocheir sinensis*) is highly sought-after by Chinese consumers due to its high nutritional value and flavorful meat [1]. Chinese mitten crab farming has recently become an important pillar of China’s fishery industry [2]. However, since 2015, hepatopancreatic necrosis disease (HPND) has become prevalent in many provinces in China, causing huge losses to the Chinese mitten crab farming industry [3]. Sick crabs exhibited slow movement and decreased appetite [4], and anatomical analyses confirmed the occurrence of hepatopancreas lesions, muscle atrophy, luminal hydrops, and gastrointestinal emptying [5,6], which was consistent with our observations (Figure 1). Although HPND-affected crabs do not die immediately, sick crabs are no longer edible, and therefore cannot be marketed.

In a previous study, HPND was first proposed to be caused by a microsporidian parasite (*Hepatospora eriocheir*) [7]. However, subsequent tests found that diseased crabs could not always detect *H. eriocheir*, and the symptoms of HPND could not be reliably reproduced via artificial *H. eriocheir* infection [8]. Interestingly, however, the injection of diseased crab supernatant (sterile) into healthy crabs, as well as animal regression experiments with dominant bacteria in the hemolymph of diseased crabs, also failed to produce HPND symptoms in healthy crabs [5,9]. This suggests that a single factor of *H. eriocheir*, bacteria, and viruses does not directly cause HPND. This was also confirmed by one of our previous meta-transcriptomics analyses, in which the viral and microsporidial communities in the hepatopancreas of HPND-affected and healthy crabs were not significantly different [3]. In general, abiotic factors are now more recognized causative factors (e.g., pesticides, high pH, and hypoxia) [10,11]. For example, Pan et al., [5] found that crabs exhibited HPND symptoms in high-pH (9.5–10) water environments. The hepatopancreas of *E. sinensis* was damaged and turned whitish under long-term exposure to low pesticide concentrations [12,13]. However, this still does not prove that HPND is caused by abiotic factors. In aquatic animals, the same disease may have different symptoms, and the same symptoms may be caused by different diseases [14]. Moreover, it is also possible that abiotic factors are just one of the causes of HPND. Therefore, additional studies are needed to gain further insights into the molecular responses and pathogenesis of HPND-affected crabs.

The recent development of effective “omic” data analysis and bioinformatic tools enables the study of the biological responses of aquatic animal cells or organs to environmental stress and disease infection through the analysis of transcriptomes, metabolomes, and proteomes [15,16]. Among them, transcriptome analysis can identify stress-induced genes and provides important information on gene expression levels, thus enabling the visualization of the expression patterns of key genes [17]. Metabolomics is the systematic study of small-molecule metabolite profiles, which describe the physiological state of an organism based on the true metabolic level in the organism [18]. Therefore, the combined analysis of transcriptomics and metabolomics may provide a deeper understanding of biological processes than either approach alone [19]. Here, comprehensive transcriptomic and metabolomic analyses revealed global changes in genes and metabolites in *E. sinensis* with HPND, thus providing insights into the adaptive mechanisms and possible pathogenic factors of diseased crabs at the molecular level.

## 2. Materials and Methods

### 2.1. Experimental Samples

The progression of the HPND is relatively long, and the hepatopancreas of sick crabs gradually degenerates and shrinks. According to the color change of the hepatopancreas, it can be divided into mild symptoms (hepatopancreas were pale yellow) and severe symptoms (hepatopancreases were milky white or grayish white) [5]. In this study, healthy crabs and severe symptomatic crabs of the same size (weight: 72.16 ± 4.02 g) were simultaneously collected from adjacent ponds of the same farm in Anfeng Town, Xinghua city, Jiangsu Province, China, to ensure that growth environment and dietary conditions were consistent. The hepatopancreases of diseased crabs (hereinafter referred to as the HPND group) and healthy crabs (control group) were rapidly resected under sterile conditions and frozen in liquid nitrogen for subsequent transcriptome sequencing and LC-MS analysis. Among them, 4 diseased crabs and 4 healthy crabs were used for transcriptome sequencing, and 10 diseased crabs and 10 healthy crabs were used for LC-MS analysis. Additionally, approximately 6 mm^3^ of tissues were dissected from the hepatopancreas and appendicular muscles of healthy and diseased crabs, then fixed in a 4% paraformaldehyde solution and stored at −4 °C after remaining at room temperature for two hours.

### 2.2. Histological Analysis

The fixed tissue was dehydrated with gradients of ethanol and then embedded in paraffin. The wax was cut into 50 μm tissue sections and stained with hematoxylin-eosin. The tissue sections were examined using a light microscope (DM6 M LIBS, Leica, Wetzlar, Germany).

### 2.3. Metabolite Extraction and Profiling Analysis

The hepatopancreas tissue ground with liquid nitrogen was vortexed with a pre-chilled 80% methanol and 0.1% formic acid solution. The sample was incubated on ice for 5 min, and then centrifuged at 15,000× *g* at 4 °C for 20 min. A certain amount of supernatant was diluted with LC-MS-grade water to a methanol content of 53%. The sample was then transferred to a new Eppendorf tube and centrifuged at 15,000× *g* at 4 °C for 20 min. Finally, the supernatant was collected and injected into an LC-MS/MS system for analysis [20]. Quality control (QC) samples were prepared by mixing equal amounts of supernatant to ensure assay stability and sample reproducibility.

UHPLC-MS/MS analysis was performed at Novogene Co., Ltd. (Beijing, China) using a Vanquish UHPLC system (Thermo Fisher, Bremen, Germany) and an Orbitrap Q ExactiveTM HF mass spectrometer (Thermo Fisher, Germany). A 17 min linear gradient was used to inject the sample into a Hypesil Gold column (100 × 2.1 mm, 1.9 μm) at a 0.2 mL/min flow rate. The measurements were conducted using eluent A (aqueous solution containing 0.1% FA) and eluent B (methanol) in positive polarity mode. Eluent A (5 mM ammonium acetate, pH 9.0) and eluent B (methanol) were used in negative polarity mode. The solvent gradient settings were as follows: 2% B, 1.5 min; 2–100% B, 12.0 min; 100% B, 14.0 min; 100–2% B, 14.1 min; 2% B, 17 min. The Q ExactiveTM HF mass spectrometer was operated in positive/negative polarity mode, with a spray voltage of 3.2 kV, a capillary temperature of 320 °C, a sheath gas flow rate of 40 arb, and an auxiliary gas flow rate of 10 arb. There were 3 QC samples in this project, and metabolites with a coefficient of variance (CV) below 30% in the QC samples were reserved as the final identification result.

### 2.4. Metabolomics Data Analysis

The raw data files generated by UHPLC-MS/MS were processed with Compound Discoverer 3.1 (CD3.1, Thermo Fisher) for peak alignment, peak detection, and quantification of each metabolite. Afterward, the peak intensity was normalized to the total spectral intensity. Normalized data were used to predict molecular formulas based on additional ions, molecular ion peaks, and fragment ions. Statistical analyses were conducted using R (R version R-3.4.3, Ross Ihaka and Robert Gentleman, New Zealand), Python (Python version 2.7.6, Guido van Rossum, USA), and CentOS (CentOS version 6.6, Gregory Kurtzer, USA). Metabolites were annotated using the KEGG (https://www.genome.jp/kegg/pathway.html, accessed on 20 November 2020), HMDB (https://hmdb.ca/metabolites, accessed on 20 November 2020), and LIPIDMaps (http://www.lipidmaps.org/, accessed on 20 November 2020) databases. Principal component analysis (PCA) and partial least squares discriminant analysis (PLS-DA) were performed in metaX to ensure the repeatability and reliability of within-group data by finding and excluding outliers in each group of samples [21]. A univariate analysis (*t*-test) was used to calculate statistical significance (*p*-value). The metabolites with VIP > 1 and *p*-value < 0.05 and fold change (FC) ≥ 2 or FC ≤ 0.5 were considered differentially expressed metabolites (DMs).

### 2.5. RNA Extraction and RNA-Sequencing

RNA was extracted from the samples using the TRIzol Reagent (Invitrogen, Carlsbad, CA, USA). RNA integrity was analyzed by 1% agarose gel electrophoresis, RNA purity was assessed with a NanoDrop spectrophotometer (Thermo Scientific, Wilmington, NC, USA), and RNA quality was confirmed with an Agilent Bioanalyzer 2100 system (Santa Clara, CA, USA). The cDNA libraries were constructed using the MGIEasy RNA Library Prep Kit. Finally, the libraries were sequenced on the MGISEQ-2000RS platform (BGI-Shenzhen, China).

### 2.6. Transcriptome Profile Analysis

The raw data were filtered using the SOAPnuke (v1.5.2, BGI-Shenzhen, China) filtering software to obtain clean data. Next, the clean reads were aligned to the reference gene sequence with Bowtie2 (v2.2.5) [22], after which the expression levels of genes and transcripts were calculated using RSEM (v1.2.12) [23]. De novo assembly of the clean reads was conducted using the Trinity suite, followed by Tgicl to cluster the assembled transcripts to de-redundancy and obtain UniGene identifiers. The obtained UniGene identifiers were then uploaded to the following main databases: KEGG (Kyoto Encyclopedia of Genes and Genomes), GO (Gene Ontology), NR (NCBI non-redundant protein sequences), NT (NCBI nucleotide sequences), SwissProt (a manually annotated and reviewed protein sequence database), and PFAM (Protein family). Afterward, gene expression was normalized to fragments per kilobase per million (FPKM) values. The DEseq2 method based on the principle of negative binomial distribution was used to detect differentially expressed genes. Genes with *p*-values < 0.05 were considered differentially expressed genes (DEGs). Next, GO and KEGG pathway enrichment analysis of these DEGs was conducted. *p*-values were FDR (false discovery rate) corrected to obtain q-values, and GO terms and KEGG pathways with q-value ≤ 0.05 were considered significantly enriched.

### 2.7. Integrative Analysis of Metabolomics and Transcriptomics

An integrative metabolomics and transcriptomics approach was adopted to better characterize the post-transcriptional regulation of gene expression. The correlation between the DEGs and the DMs was determined using the Pearson statistical method and the correlation coefficient R^2^ and *p*-value of the DEGs and DMs were calculated. All of the differentially expressed genes and metabolites were mapped to the KEGG pathway database, their common pathway information was obtained, and a KEGG enrichment bubble chart was generated for the pathways that were significantly enriched with DEGs and DMs using the ‘ggplot2’ R package.

## 3. Results

### 3.1. Histopathology of Crab Tissues

As shown in Figure 1, the hepatopancreas and muscles of crabs with HPND exhibited tissue damage compared with healthy tissues. In the hepatopancreas, the hepatic tubules were severely damaged, and a large number of vacuoles was observed. (Figure 1E). Additionally, muscle fibers appeared loose or broken and nuclei became smaller (Figure 1F).

### 3.2. Transcriptomic Analysis of E. sinensis with HPND

After de-redundancy by clustering the assembled transcripts, 80,869 UniGenes were obtained. The total length, average length, N50, and GC content were 73,433,519 bp, 908 bp, 1668 bp, and 43.40%, respectively. The UniGenes were then compared to seven functional databases for annotation, and functional annotations were obtained for 33,762 (NR: 41.75%), 30,982 (NT: 38.31%), 21,769 (SwissProt: 26.92%), 20,610 (KOG: 25.49%), 23,392 (KEGG: 28.93%), 20,946 (GO: 25.90%), and 18,252 (Pfam: 22.57%) UniGene identifiers.

A total of 3332 DEGs were identified in the HPND vs. control group, of which 1491 DEGs were significantly up-regulated and 1841 DEGs were significantly down-regulated (Figure 2). These DEGs were subjected to GO and KEGG functional analysis to better understand the biological significance of these DEGs and the biochemical processes involved. The top 20 GO enrichment terms indicated that the DEGs were mainly involved in some important biological processes and molecular functions, such as cellulase activity, viral RNA genome replication, cellulose catabolic process, RNA-directed 5′-3′ RNA polymerase activity, and vacuolar transport. Moreover, the results of KEGG enrichment show that some pathways, such as the IL-17 signaling pathway, lysosome, melanogenesis, protein digestion and absorption, starch and sucrose metabolism, and thiamine metabolism underwent significant changes (Figure 3).

### 3.3. Metabolic Analysis of E. sinensis with HPND

LC-MS detections were performed in positive-ion (POS) modes and negative-ion (NEG) modes to maximize metabolite coverage. In the PCA analysis, the samples of different groups were grouped in distinct clusters, and the QC samples were clustered together, indicating that the samples of different groups were quite different, and the experiment was reproducible (Appendix A). Additionally, the PLS-DA model of each comparison group was established. The model evaluation parameters (POS: R2Y = 0.98, Q2Y = 0.92, NEG: R2Y = 0.97, Q2Y = 0.91) were obtained through 7-fold cross-validation, indicating that the model was well-established (Figure 4A,B). Next, the PLS-DA model was further validated by 200 permutation experiments. All R2 data were larger than the Q2 data in the positive and anion modes, and the Q2 regression line with Y-intercept was less than 0, indicating that the model was robust and reliable without overfitting (Figure 4C,D). Therefore, the experimental instrument analysis system was considered stable, and the data were reliable.

We screened for metabolites that were significantly different between the two groups based on the following criteria: VIP > 1.0, FC > 1.5 or FC < 0.667, and *p*-value < 0.05. In the POS and NEG mode, a total of 554 DMs were identified in the HPND vs. control group. Among them, 365 metabolites were in POS mode (210 up-regulated, 155 down-regulated) and 189 metabolites were in NEG mode (140 up-regulated, 49 down-regulated). In these DMs, a total of 87 metabolites were related to lipids, of which fatty acids (34) and glycerophospholipids (37) were the most abundant. KEGG enrichment analysis was performed on these DMs to explore the most relevant pathways. The results show that the top 20 enriched pathways were mainly related to the cGMP-PKG signaling pathway, purine metabolism, pyrimidine metabolism, arachidonic acid metabolism, amino acid biosynthesis, and neuroactive ligand–receptor interactions (Figure 5).

### 3.4. Integrated Analysis of Metabolome and Transcriptome

To further screen related genes and metabolites, as well as key metabolic pathways, we performed a comprehensive analysis of the transcriptome and metabolome. As shown in Appendix A, the top 50 differential metabolites had strong correlations with the top 100 differential genes. The KEGG enrichment results of DMs and DEGs identified several important response pathways in *E. sinensis* with HPND, including protein digestion and absorption, neuroactive ligand–receptor interactions, lysosome, purine metabolism, pyrimidine metabolism, oxidative phosphorylation, TCA cycle, and cholinergic synapse (Figure 6).

### 3.5. Molecular Responses of E. sinensis with HPND

Based on the results of comprehensive transcriptome and metabolome analysis, we found that the immune system, metabolic capacity, antioxidant system, detoxification capacity, and nervous system of Chinese mitten crabs with HPND were affected. Figure 7 shows hypothetical pathways for molecular responses in diseased crabs. Interestingly, many genes related to autophagy and apoptosis were significantly up-regulated in the HPND group, which may suggest that higher levels of cellular degradation occurred in the hepatopancreas of HPND-affected crabs. More details are provided in Appendix A. Some metabolites, such as taurine, arachidonic acid, uracil, xanthine, adenine, and uridine are associated with the immune system and were dysregulated in our study (Table 1). Additionally, we found that some genes and metabolites that participate in mitochondrial function were dysregulated, most of which were up-regulated. For example, in the mitochondrial respiratory chain complex I (NADH dehydrogenase), the ND4 and ND5 genes were significantly down-regulated, and Ndufab12 was significantly up-regulated; COX7A, COX11, and COX15 were significantly up-regulated in complex IV (cytochrome c oxidase); and ATP6F and ATP6C were significantly up-regulated in complex V (ATP synthase).

## 4. Discussion

Due to its high fatality rate (40–50%), the etiology and pathogenesis of HPND have attracted much attention [7]. Previous studies have shown that a single factor of pathogens, bacteria, and viruses does not directly cause HPND. After investigating the living environment, offspring seeds, and epidemiology of Chinese mitten crabs, recent studies have inferred that HPND is likely caused by abiotic factors. However, the real pathogenic factors still need to be explored further [3,4,5]. In this study, transcript and metabolite changes in diseased crabs were determined by transcriptomics and a metabolomics analysis of healthy and HPND-affected crabs.

### 4.1. Abnormalities of the Nervous System

In invertebrates, the nervous system plays a key role in coping with stress. Neurons can not only receive and process cues from environmental stressors but can also actively respond to promote survival [46,47]. Nonetheless, environmental stress can cause irreversible damage to neurons [48]. For example, hypoxia can cause neuronal dysfunction in *C. elegans*, leading to irreversible neuronal damage [47]. In this study, we found that some differentially expressed genes and metabolites were enriched in the calcium signaling pathway, neuroactive ligand–receptor interaction, and cholinergic synapse.

It has been reported that some pesticides (e.g., trichlorfon, deltamethrin, and chlorpyrifos) can cause neurotoxicity in Chinese mitten crab, which in turn affects the transmission of nerve impulses [12,49]. The systemic toxicity of these pesticides lies in their ability to inhibit acetylcholinesterase (AChE) [50,51]. The main biological role of AChE is to rapidly hydrolyze the neurotransmitter acetylcholine (ACh) to terminate impulse transmission at cholinergic synapses [44]. Nerve conduction becomes inhibited when ACh accumulates excessively, resulting in neuronal dysfunction and death [45]. Therefore, the inhibition of AChE activity is often used as a biomarker of neurotoxic substances [52]. In our study, the activity of AChE was inhibited, and the content of acetylcholine increased. This suggests that the diseased crabs may have been affected by a neurotoxin. Additionally, the content of L-glutamate was significantly up-regulated in the HPND group. L-glutamate is an important neurotransmitter and can activate several subtypes of receptors, resulting in increased intracellular calcium concentration [43,49]. When calcium levels rise in postsynaptic neurons, a cascade of signaling events is activated, which is critical for neuronal signal processing and survival [53]. Furthermore, the activation of neuroactive ligand–receptor interactions is important for neural repair [54]. Therefore, our findings suggest that failure to detect and adapt to environmental stressors can result in irreversible neuronal damage in diseased crabs.

### 4.2. Oxidative Stress and Impaired Detoxification

Studies have shown that intense environmental stress can disrupt the homeostasis of reactive oxygen species (ROS) in aquatic organisms, resulting in elevated levels of ROS and damage to cellular components, which is commonly referred to as “oxidative stress” [55,56]. Among the damaged cellular components are macromolecules, such as lipids and proteins [57]. Therefore, heat shock proteins (HSPs) can prevent cellular oxidative stress by renaturing denatured proteins [58]. In particular, HSP70 and HSP90 are often used as indicators of the health status of aquatic organisms [59,60]. Specifically, HSP70 can maintain cellular homeostasis, and HSP90 can prevent irreversible protein aggregation under stressful conditions [36,61]. Therefore, these proteins are typically up-regulated in Chinese mitten crabs under oxidative stress, as demonstrated in studies of crabs exposed to imidacloprid and avermectin [62,63]. Thus, the up-regulation of HSP70 and HSP90 in the HPND group in our study is likely a reflection of the body’s oxidative stress state.

Additionally, our study also found significant changes in two antioxidants PRDX5 and Glutathione S-transferase (GST). PRDX5 is a catalase that protects glial cells from oxidative stress [34]. The overexpression of PRDX5 further confirmed that the nervous system of the diseased crab may be affected. In contrast, the expression of GST was significantly inhibited, which may be related to the detoxification effect of GST. GST and carboxylesterase (CarEs) are two important detoxification enzymes in insects and play an important role in pesticide metabolism [64]. In crustaceans, these two enzymes are primarily active and metabolized in the hepatopancreas [65]. Among them, GST is a group of transferase enzymes involved in pesticide metabolism, and its activity is related to pesticide metabolism [38]. CarEs was identified as the main enzyme for decomposing pyrethroids and can be used as an indicator of pyrethroid toxicity [39]. The activities of CarEs and GST were found to be inhibited in many experiments on pesticide stress in Chinese mitten crabs [12,13]. Therefore, the expression of these two enzymes was down-regulated in the HPND group, indicating that the diseased crabs were likely to be stressed by pesticides such as pyrethroids, resulting in a decrease in their detoxification ability. In our metabolome data, the presence of some toxic phenylpropanoids and polyketides, such as 4-methoxycinnamic acid, aflatoxin G1, and 3-hydroxy-4-methoxycinnamic acid, further support this hypothesis.

However, we did not find significant changes in the activities of other antioxidants such as superoxide dismutase (SOD) and catalase (CAT), which are often used as indicators of oxidative stress in environmental pollution studies [66]. One possible explanation is that Chinese mitten crabs with HPND are exposed to lower concentrations of environmental pollutants. For example, Hong et al., found that when Chinese mitten crabs were exposed to less than 4.4 mg/L glyphosate, and the activities of SOD and peroxidase (POD) did not significantly change [67]. Collectively, our findings suggest that diseased crabs may be stressed by environmental pollutants, resulting in oxidative stress and impaired detoxification.

### 4.3. Increase in Autophagy and Apoptosis

Hepatopancreas damage is the most relevant pathological feature of *E. sinensis* with HPND (Figure 1). Previous studies found that hepatocytes from diseased crabs exhibit nuclear chromatin condensation, endoplasmic reticulum disintegration, and mitochondrial and lipid droplet cristae deformation. This is a pathological feature of apoptosis and autophagy [5]. Additionally, liver injury is reported to be closely related to autophagy and apoptosis [68,69]. In our study, many genes related to autophagy and apoptosis were found to be altered, and most of them were up-regulated.

To survive starvation and other forms of stress, eukaryotic cells can degrade the cytoplasm through lysosomes and this process is generally referred to as autophagy [70]. During the early stages of autophagy, Beclin-1 can activate downstream genes by recruiting other autophagy-related (Atg) proteins [71]. For example, the Beclin 1-Atg14L complex may participate in the early steps of autophagic biosynthesis by recruiting Atg5 and Atg12 [24]. Therefore, the overexpression of Atg14L can promote the formation of autophagosomes and up-regulate autophagy [24]. In our study, Atg14L was significantly up-regulated in the HPND group, as well as Atg1 and Atg7. The overexpression of Atg1 was shown to be sufficient to induce high levels of autophagy [26]. Atg7 plays an important role in starvation-induced autophagy [25]. Therefore, our study suggests that high levels of autophagy occur in diseased crabs. The overexpression of Atg7 may be associated with the emergence of the jejunum of diseased crabs. Rubicon is a Beclin 1-interacting protein whose forced expression results in abnormal lysosomal structure and impaired autophagosome maturation, thus inhibiting autophagy [24]. Interestingly, we found that Rubicon was significantly up-regulated and lysosomal-related genes and metabolites were significantly induced in the HPND group, which may be a manifestation of uncontrolled autophagy in the body.

Apoptosis is a form of programmed cell death that can be mediated by mitochondria, the endoplasmic reticulum, and death receptors [72,73]. In eukaryotes, an increase in misfolded or unfolded proteins causes a stress response in the endoplasmic reticulum, which reduces protein synthesis and increases protein folding [30]. However, excessive or prolonged ER stress can lead to apoptosis [74]. PERK and IP3R play important roles in endoplasmic reticulum stress-induced apoptosis. PERK is a protein kinase distributed on the endoplasmic reticulum membrane. When high-intensity or prolonged endoplasmic reticulum stress occurs, PERK will activate the transcriptional expression of the transcription factor ATF4, thereby promoting the expression of the apoptosis signaling molecule CHOP, which in turn promotes apoptosis [29,75]. On the other hand, a large number of calcium ions in the endoplasmic reticulum can enter cells through the activated IP3R channel and activate Caspase-4, thereby affecting apoptosis [76,77]. Therefore, the significant up-regulation of PERK, ATF4, CHOP, and Caspase-4 in our study suggested the activation of ER-mediated apoptosis in diseased crabs. Based on previous results, this may be related to oxidative damage in the diseased crabs. In mitochondria-mediated apoptosis, mitochondria can release cytochrome c by changing the outer membrane permeability (MOMP) and gradually activate Apaf-1 and Caspase-9 with the assistance of ATP and dATP. These components then form apoptotic bodies to activate caspase-3 and caspase-7, resulting in apoptosis [31,78]. Therefore, Bcl-2 family proteins can promote or inhibit apoptosis by changing MOMP and then affecting the release of cytochrome c [79]. In our study, ADP content was significantly increased and Caspase-7 expression was significantly up-regulated. However, cytochrome c and Caspase-9 did not significantly change, which may be related to the significant up-regulation of BNIP3 and BUFFY. BNIP3, a hypoxia-induced cell death protein, can cause mitochondrial dysfunction [27,80]. BUFFY is a Bcl-2 family protein with anti-apoptotic effects and its overexpression also induces cell death [28,81]. Therefore, our study indicated that diseased crabs presented high levels of apoptosis, as well as the possibility of abnormal mitochondrial function. It is worth mentioning that diseased crabs die within a short period of time out of the water. Previous studies have shown that hypoxia exacerbates the occurrence of HPND [11]. We speculate that this may be related to the overexpression of BNIP3 and mitochondrial dysfunction.

IL-17 is known to have pro-inflammatory effects and is associated with a variety of autoimmune diseases [82,83]. However, studies have shown that activation of the IL-17 signaling pathway is also closely related to autophagy and apoptosis [84,85]. For example, IL-17-induced apoptosis was alleviated after Act1 knockdown [86]. Furthermore, Zhao et al. found that fluoride could induce apoptosis and autophagy by activating the IL-17 signaling pathway and ultimately damage hepatocytes [87]. The present study also found that the IL-17 signaling pathway was activated in the diseased crabs. This may suggest that the hepatopancreas damage of the Chinese mitten crabs with HPND may be the result of excessive autophagy and apoptosis induced by environmental pollutants. Based on our previous results, we suspect that this may be related to the recent and widespread use of pyrethroids for pond cleaning in Xinghua city, Jiangsu province.

### 4.4. Impairment of Immune System

As an invertebrate, the Chinese mitten crab lacks adaptive immunity and only relies on innate immunity. In this study, we found that in addition to the lysosome and IL-17 signaling pathways, other pathways related to innate immunity were significantly induced in the diseased crabs, such as the CGMP-PKG signaling pathway, melanogenesis, and endocytosis. Crustacean innate defense systems include cellular and humoral immunity [88]. In humoral immunity, lysosomes can degrade substances through their own various hydrolases, thus playing an important role [89]. Alkaline phosphatase (AKP), acid phosphatase (ACP), lysozyme (LZM), cathepsin L, and cathepsin B are several important hydrolases in lysosomes. Among them, AKP, ACP, and LZM are considered to be the first line of defense within the humoral defense system [32]. Cathepsin L and cathepsin B are important proteases in lysosomes [90]. In our study, genes other than ACP were significantly induced. The insignificant changes in ACP may be related to the fact that a high pH is suspected to induce the occurrence of HPND [5]. Phagocytosis and apoptosis are important cellular immune response processes in invertebrates. Lectin and cathepsin L contribute to phagocytosis [33]. In our study, the C-type lectin, C-type lectin receptor protein, and cathepsin L genes were significantly down-regulated in diseased crabs, suggesting that the phagocytic activity of diseased crabs was reduced. Therefore, we concluded that the innate immunity of Chinese mitten crabs with HPND would be affected. It is worth mentioning that the high incidence of HPND is from May to July each year, during which time the weather changes greatly and the temperature is high. Dramatic changes in climate can further weaken a crustacean’s immune system and exacerbate the development of the disease [91]. Therefore, we suspect that unstable weather may exacerbate the occurrence of HPND.

### 4.5. Energy and Substance Metabolism Disorders

In response to environmental challenges, aquatic organisms can maintain homeostasis by increasing energy production, such as up-regulating the TCA cycle, oxidative phosphorylation, and glycolysis [15,48,92]. Pyruvate is the end product of glycolysis, and the resulting pyruvate can be converted into acetyl-CoA to enter the TCA cycle. In the HPND group, we found that some key metabolites (dihydroxyacetone phosphate, succinate, citrate) and key genes (hexokinase, acetyl-CoA synthetase, aconitate hydratase) involved in glycolysis and the TCA cycle were significantly up-regulated. The same result also occurs in oxidative phosphorylation (Figure 7). This indicates that Chinese mitten crabs with HPND appropriately increased their energy production. It is also worth mentioning that anaerobic metabolism is characterized by high levels of end products (e.g., alanine and succinate) in invertebrate species [40,93]. Therefore, the accumulation of succinate in the HPND group in our study is likely to indicate the occurrence of anaerobic metabolism in diseased crabs.

Lipids play an important role in energy storage and release in organisms. In our metabolomic data, many metabolites were associated with lipids, with fatty acids and glycerophospholipids accounting for the largest proportions of lipids. Studies have shown that fatty acid metabolism in crustaceans is closely related to immune responses [94]. As an essential fatty acid, arachidonic acid can affect the activity of cell membranes and membrane-associated enzymes, thereby altering immune function [41]. Abnormal arachidonic acid metabolism was also found in our results. Therefore, abnormal fatty acid metabolism may be related to immune dysregulation in diseased crabs. It is generally known that biological membranes are mainly composed of phospholipids. Most pesticides are highly lipophilic, and thus they primarily affect cell membranes [95]. Interestingly, almost all glycerophospholipid metabolites were downregulated in our results (Appendix A).

Furthermore, some differentially expressed genes and metabolites were significantly enriched in pyrimidine and purine metabolism pathways in our results. Pyrimidines and purines are essential for life and perform many important cellular functions [96]. Viral infection can alter some metabolites involved in pyrimidine and purine metabolism [97,98]. Likewise, the apoptosis of hepatopancreatic cells can lead to disturbances in nucleotide metabolism [99]. Therefore, disturbances in pyrimidine and purine metabolism may lead to immunodeficiency [100]. In our study, some metabolites in purine and pyrimidine metabolic pathways, such as uracil, xanthine, adenine, and uridine, were significantly up-regulated in the diseased crabs. The accumulation of uracil and xanthine has been reported to induce apoptosis [42]. Therefore, we could deduce that the dysregulation of purine and pyrimidine metabolism in diseased Chinese mitten crabs is related to the high levels of autophagy and apoptosis in these organisms.

Additionally, we found that several other metabolic pathways were also altered, including starch and sucrose metabolism, protein digestion and absorption, vitamin digestion and absorption, and alanine, aspartate, and glutamate metabolism. Therefore, our findings suggest that the metabolic capacity of *E. sinensis* with HPND is adversely affected.

## 5. Conclusions

In conclusion, this study provide insights into metabolic changes and functional changes of *Eriocheir sinensis* with HPND by adopting a comprehensive metabolomic and transcriptomic analysis approach. The enriched pathways of DEGs and DMs may be related to the pathological mechanism of HPND and suggest that HPND could mainly be caused by abiotic factors. However, additional studies are required to identify the true pathogenic cause of HPND and whether it is caused by a single factor or a combination of factors.

## Figures and Tables

**Figure 1 biology-11-01267-f001:**
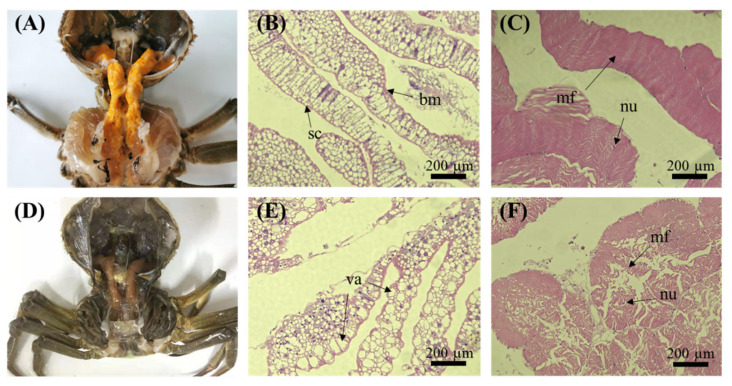
Healthy (**A**) and HPND-affected (**D**) *E. sinensis.* (**B**) and (**E**) represent the hepatic tubules of healthy and diseased crabs, respectively. (**C**) and (**F**) represent the appendage muscle tissue of healthy and diseased crabs, respectively. bm: basement membrane; nu: nucleus; mf: muscle fiber; va: vacuole; sc: secretory cell.

**Figure 2 biology-11-01267-f002:**
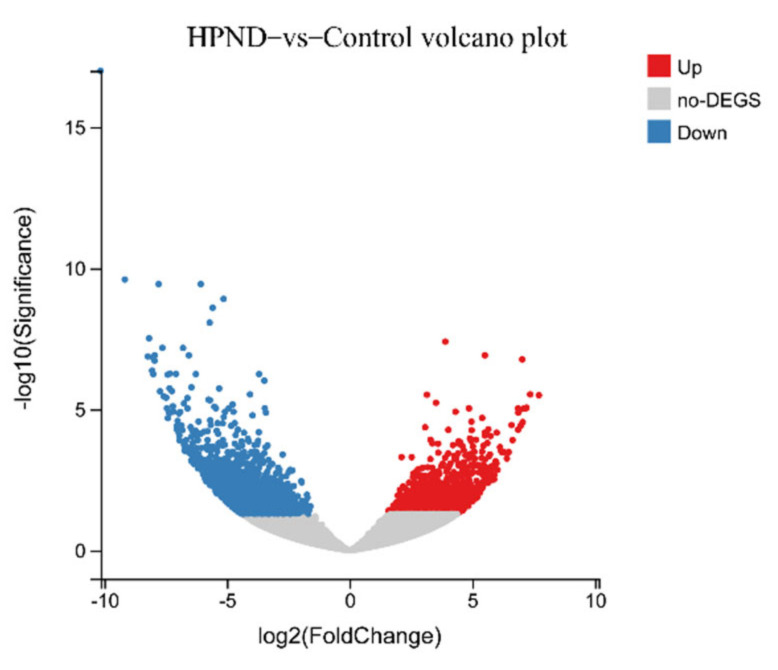
Volcano plot of differentially expressed genes.

**Figure 3 biology-11-01267-f003:**
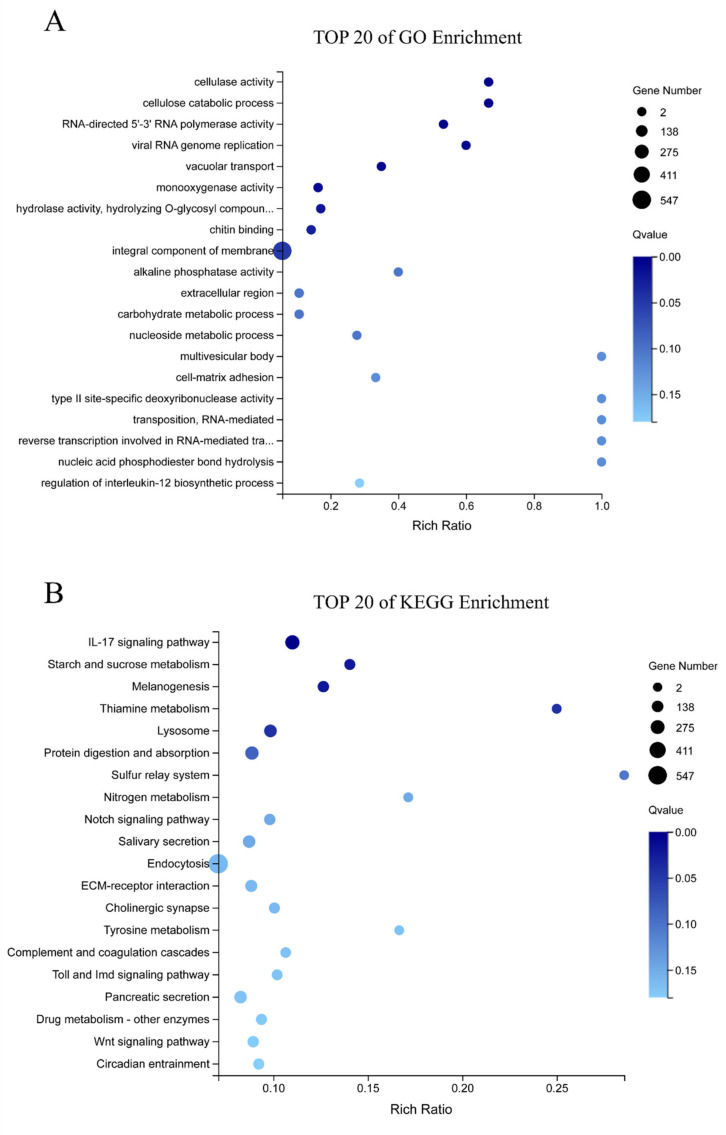
Top 20 GO and KEGG pathways significantly enriched with differentially expressed genes ((**A**) top 20 of GO enrichment; (**B**) top 20 of KEGG enrichment).

**Figure 4 biology-11-01267-f004:**
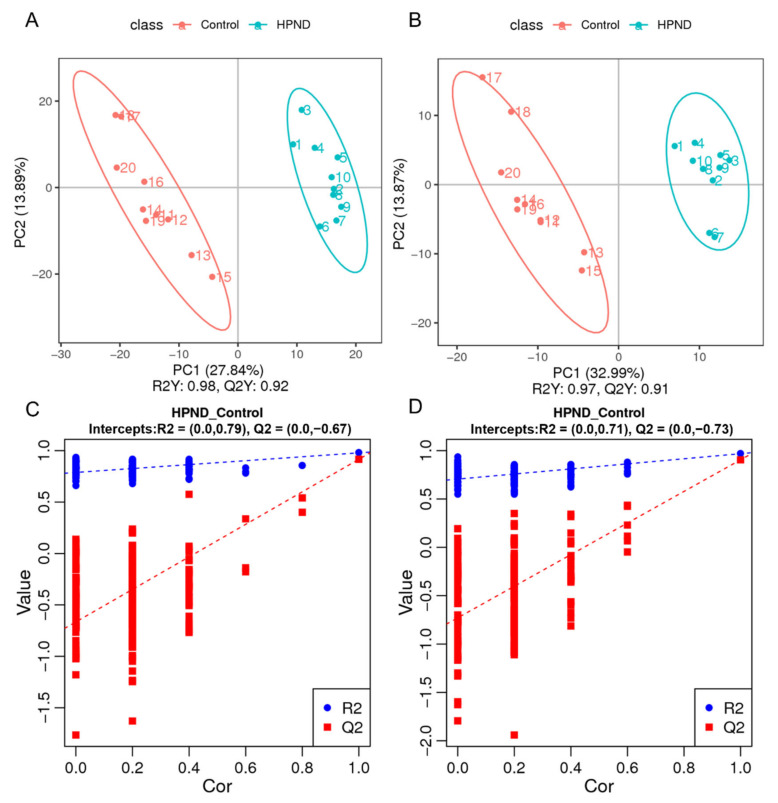
PLS-DA score plots in different groups ((**A**,**C**) POS mode; (**B**,**D**) NEG mode).

**Figure 5 biology-11-01267-f005:**
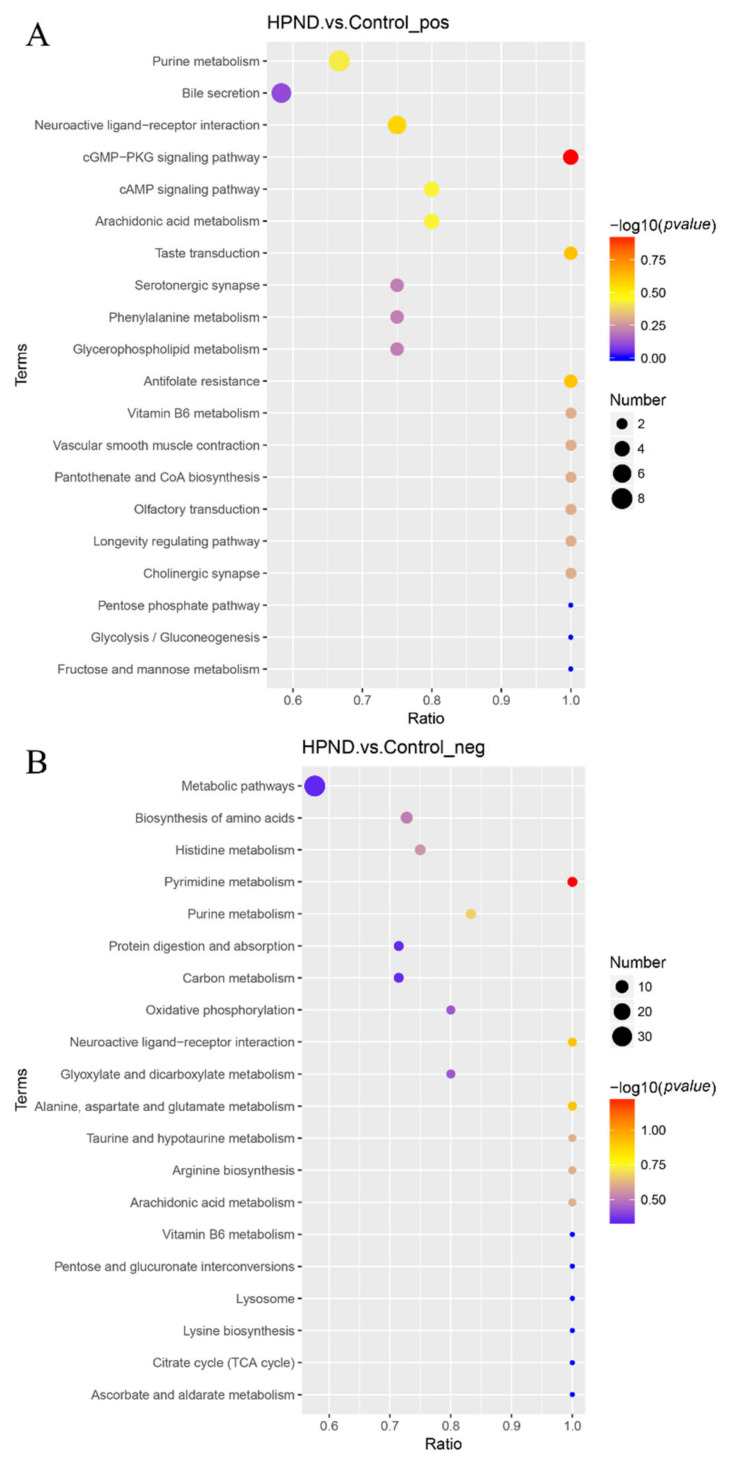
Top 20 KEGG pathways significantly enriched with differentially expressed metabolites ((**A**) POS mode; (**B**) NEG mode).

**Figure 6 biology-11-01267-f006:**
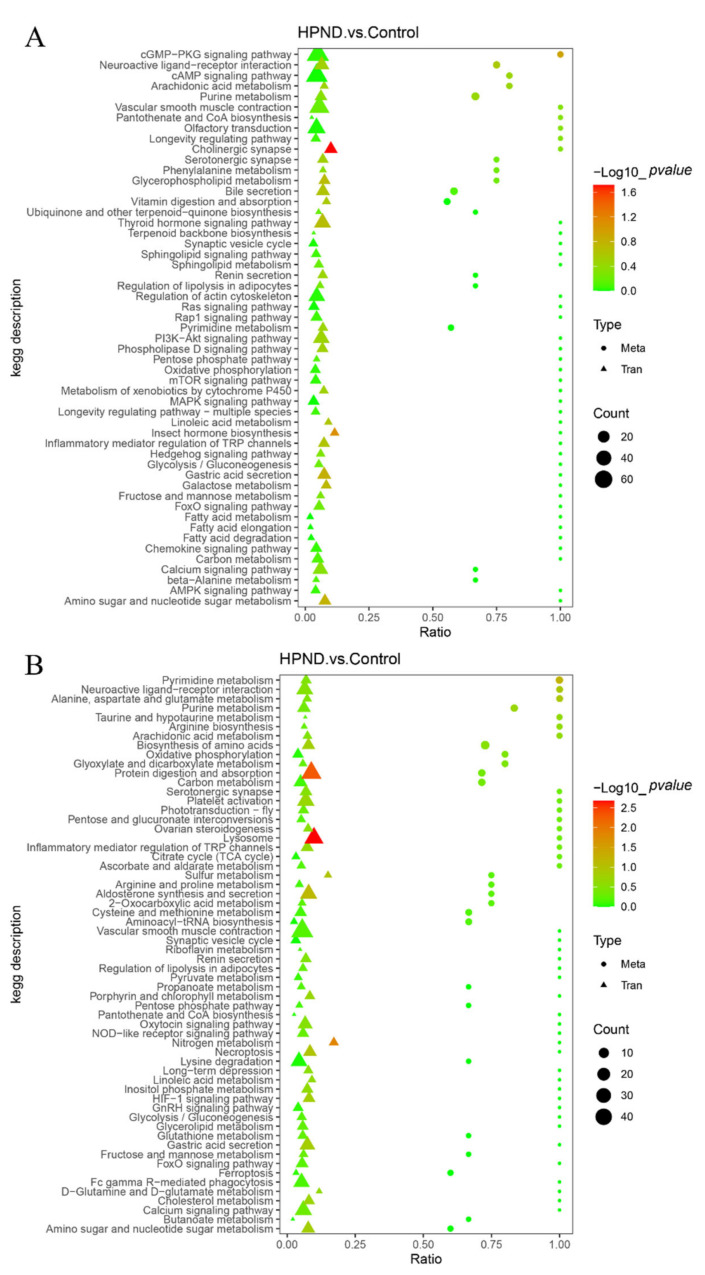
KEGG enrichment analysis for differentially expressed genes and metabolites ((**A**), POS mode; (**B**), NEG mode).

**Figure 7 biology-11-01267-f007:**
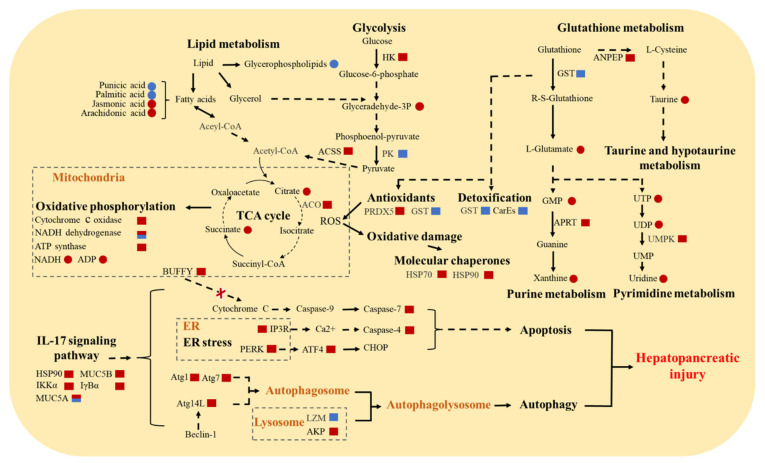
Hypothetical pathways for metabolome and transcriptome changes in *E. sinensis* with HPND. The boxes represent genes, and the dots represent metabolites. Red and blue represent up- and down-regulation, respectively, and both red and blue represents both up- and down-regulation. The solid arrows and dashed arrows indicate direct and indirect effects. Arrows with × indicate inhibition. PK: pyruvate kinase; ACSS: acetyl-CoA synthetase; ACO: aconitate hydratase; HK: hexokinase; UMPK: UMP-CMP kinase; APRT: adenine phosphoribosyltransferase; ANPEP: aminopeptidase N.

**Table 1 biology-11-01267-t001:** Significantly dysregulated genes and metabolites in crabs with HPND.

Name	Gene/Metabolite	Up/Down	Function	Reference
**Immune system**				
beclin 1-associated autophagy-related key regulator (ATG14L)	gene	up	promote the formation of autophagosomes	[24]
ubiquitin-like modifier-activating enzyme ATG7 (ATG7)	gene	up	associated with starvation-induced autophagy	[25]
threonine-protein kinase ULK2 (ATG1)	gene	up	induce high levels of autophagy	[26]
BCL2/adenovirus E1B 19 kDa protein-interacting protein 3(BNIP3)	gene	up	cell death protein	[27]
run domain Beclin-1 interacting and cysteine-rich containing protein (RUBICON)	gene	up	inhibit autophagy	[24]
Bcl-2 family protein (BUFFY)	gene	up	inhibit apoptosis	[28]
ER protein kinase (PERK)	gene	up	promote apoptosis	[29]
Inositol 1,4,5-Trisphosphate Receptor (IP3R)	gene	up	release calcium ions	[30]
caspase 7(CASP7)	gene	up	molecular triggers of apoptosis	[31]
caspase 4 (CASP4)	gene	up	molecular triggers of apoptosis	[30]
Lysozyme (LZM)	gene	down	lysosomal hydrolase	[32]
alkaline phosphatase (AKP)	gene	up	lysosomal hydrolase	[32]
C-type lectin (CTL)	gene	down	aids in phagocytosis	[33]
**Antioxidant system**				
peroxiredoxin 5(PRDX5)	gene	up	thioredoxin peroxidase	[34]
glutathione S-transferase (GST)	gene	down	removes lipid peroxides and hydrogen peroxide	[35]
heat shock protein 70 (HSP70)	gene	up	maintain cellular homeostasis	[36]
heat shock protein 90 (HSP90)	gene	up	prevents irreversible protein aggregation	[37]
**Detoxification**				
glutathione S-transferase (GST)	gene	down	decomposition of pesticides	[38]
carboxylesterase (CarEs)	gene	down	enzyme for decomposing pyrethroids	[39]
**metabolic system**				
succinate	metabolite	up	alternative end product of anaerobic metabolism	[40]
arachidonic acid	metabolite	up	involved in immune response	[41]
uracil	metabolite	up	related to apoptosis	[42]
xanthine	metabolite	up	related to apoptosis	[42]
**nervous system**				
L-glutamate	metabolite	up	neurotransmitter	[43]
acetylcholinesterase (AChE)	gene	down	hydrolyzed acetylcholine	[44]
acetylcholine (ACh)	metabolite	up	neurotransmitters	[45]

## Data Availability

The transcriptome datasets generated by this study have been deposited into the NCBI SRA database (Accession number: SUB11342700; https://submit.ncbi.nlm.nih.gov/subs/sra/SUB11342700, accessed on 17 April 2022).

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
