# Peer review of "Integrative Analysis of Transcriptome and Metabolome Reveals Molecular Responses in Eriocheir sinensis with Hepatopancreatic Necrosis Disease"

_biology, 2022, doi:10.3390/biology11091267_

Round 1

Reviewer 1 Report

I want to congratulate the authors on performing a very interesting study on Chinese mitten crab hepatopancreatic necrosis disease. This integrative transcriptomics and metabolomics study shed light on molecular responses in diseases crabs. Specifically, authors highlighted functional hypoxia as a mechanism important for the progression of diseases crabs.

 I think the methodology and data presentations was performed adequately. I have few suggestions and clarification points:

-Regarding the PCA analysis, will you consider presenting principal components other than PC1 and PC2? For example, a bar plot of percentage of top 10 PCs can be interesting?

-In Figure 7, there are purple squares which represents both up- and down-regulation. Can authors be more clear on this?

-Can you please have some discussions how important purine and pyrimidine metabolism (both in red in Fig 7)  are for the hepatopancreatic injury? Or do the authors believe the pathologic mechanism is only due to apoptosis and autophagy?

-In section 3.5, authors state some of the up- and down-regulated mitochondrial respiratory chain proteins. Is there an explanation why some of these proteins are up- and some down-regulated?

Author Response

Point 1: Regarding the PCA analysis, will you consider presenting principal components other than PC1 and PC2? For example, a bar plot of percentage of top 10 PCs can be interesting?

Response 1: Thank you for your suggestion. We regret that we didn't export the bar plot of percentage of top 10 PCs. We are very sorry for the inadequacy of our work.

Point 2: In Figure 7, there are purple squares which represents both up- and down-regulation. Can authors be more clear on this?

Response 2: Thank you for pointing this out. We have replaced the purple in Figure 7 with half red and half blue to better represent both up- and down-regulation.

Point 3: Can you please have some discussions how important purine and pyrimidine metabolism (both in red in Fig 7)  are for the hepatopancreatic injury? Or do the authors believe the pathologic mechanism is only due to apoptosis and autophagy?

Response 3: Thank you for pointing this out. According to previous studies, dysregulation of purine and pyrimidine metabolism is associated with immunodeficiency. For example, the accumulation of uracil and xanthine can induce apoptosis. Apoptosis of hepatopancreatic cells can also lead to disturbances in nucleotide metabolism. Here, we believe that apoptosis and autophagy are the most important pathological mechanisms. Therefore, we prefer that the abnormality of purine and pyrimidine metabolism is mainly due to immunodeficiency. We have made appropriate revisions to the relevant content of the article to better express our views.

Point 4: In section 3.5, authors state some of the up- and down-regulated mitochondrial respiratory chain proteins. Is there an explanation why some of these proteins are up- and some down-regulated?

Response 4: Thank you for pointing this out. In our data, most of the genes and metabolites related to the mitochondrial respiratory chain were up-regulated, and only a few were down-regulated. We suspect that this result is related to impaired mitochondrial function, which we have described in the article.

Reviewer 2 Report

Dear Editor,

The MS entitled “Integrative analysis of transcriptome and metabolome reveals molecular responses in Eriocheir sinensis with hepatopancreatic necrosis disease” provides interesting evidence of the effect of HPND in the hepatopancreas of the mitten crab, an important economic invertebrate in China.

General comments

What was the grade of infection or symptoms of the crabs for the study? the results show autophagy and apoptosis genes. How did the authors select infected organisms to have a representative picture of the same grade of disease?

The authors suggest in the discussion section that a neurotoxin could be implicated in the HPND, also line 347 suggests that may be pesticides could have a role in the disease. Did the author analyze the water of the farm? Is there any evidence of putative toxins in the farm or pesticides? 

Specific comments

Line 43: Add the scientific name after the common name. “Chinese mitten crab (Eriocheir sinensis)

Line 46: add the acronym HPND after the full name. “hepatopancreatic necrosis disease (HPND)”

Line 51: Figure 1

Line 53: Delete (Ding et al., please only year should be shown.

Histological analysis

What part of the hepatopancreas and muscle was collected for histology? The anatomy of the hepatopancreas is complex, please specify the section collected, also for the muscle.

Line 152 Size letters are different from the above subtitles.

Line 349-350. Authors claim that pesticides may be involved in HPND, and the presence of some molecules supports their hypothesis, however, did the author find the same molecules in the control group? the organisms collected are from the same pond or were farmed in a different pond? What is the rate of sick crabs found on the farm, where the organisms were collected? If so, why do some crabs get sick and others don´t?

Can you find healthy and sick crabs in the same area?

The paragraph of the 4.2 section is too long, please separate it for better reading.

Conclusion

Authors need to rephrase the conclusions in one general conclusion, instead of summarizing each analysis again. Authors claim that environmental stress, pesticides, and abiotic factors may be involved in HPND.

Figures

Figure 1. Please correct units, it should be µm instead of um, and separate numbers from the units.

Figures 2, 3, 5, and 6. Figures are not readable, please modify them to increase quality. These figures could be saved as TIFF directly from R software.

Figure 3. Please add A) to the first figure and B) to the second in the figure legend. Also, it could be better to arrange figure 3 vertical format.

Figure 6 could be replaced by a heatmap. Triangles and spheres are in black in the reference legend, however, the figure is green to red. Graphs could be arranged vertically for better presentation.

Figure 7. Check the word in the figure, several typos.

Example “Molocular”

Author Response

Point 1: What was the grade of infection or symptoms of the crabs for the study? the results show autophagy and apoptosis genes. How did the authors select infected organisms to have a representative picture of the same grade of disease?

Response 1: Thank you for pointing this out. Crabs with HPND can be divided into mild symptoms (the color of the hepatopancreas is pale yellow) and severe symptoms (the color of the hepatopancreas is milky white or grayish white) according to the degree of damage to the hepatopancreas. Crabs with severe symptoms were used in this study, which have been annotated in the article.

Point 2: The authors suggest in the discussion section that a neurotoxin could be implicated in the HPND, also line 347 suggests that may be pesticides could have a role in the disease. Did the author analyze the water of the farm? Is there any evidence of putative toxins in the farm or pesticides? 

Response 2: We are profoundly thankful to the reviewers for their insights and professionalism. We regret that we do not test the water on the farm and have no evidence of the putative toxins and pesticides on the farm. We can only infer from our data results that the diseased crabs may have been stressed by abiotic factors such as pesticides. We are sorry for the inadequacies in our work.

Point 3: Line 43: Add the scientific name after the common name. “Chinese mitten crab (Eriocheir sinensis)

Response 3: Thank you for pointing this out. We have added the scientific name after the common name.

Point 4: Line 46: Add the acronym HPND after the full name. “hepatopancreatic necrosis disease (HPND)”

Response 4: Thank you for pointing this out. We have added the acronym HPND after the full name.

Point 5: Line 51: Figure 1

Response 5: Thank you for pointing this out. We have made changes to this.

Point 6: Line 53: Delete (Ding et al., please only year should be shown.

Response 6: Thank you for pointing this out. We have delete (Ding et al..

Point 7: What part of the hepatopancreas and muscle was collected for histology? The anatomy of the hepatopancreas is complex, please specify the section collected, also for the muscle.

Response 7: Thank you for pointing this out. The hepatopancreas of Eriocheir sinensis has basically the same structure and is composed of many branched hepatic tubules. In this study, the tissue section of the hepatopancreas shows the hepatic tubules, and the muscle tissue section shows the appendicular muscle tissue. We have explained it in the article.

Point 8: Line 152 Size letters are different from the above subtitles.

Response 8: Thank you for pointing this out. We have made changes in the article.

Point 9: Line 349-350. Authors claim that pesticides may be involved in HPND, and the presence of some molecules supports their hypothesis, however, did the author find the same molecules in the control group? the organisms collected are from the same pond or were farmed in a different pond? What is the rate of sick crabs found on the farm, where the organisms were collected? If so, why do some crabs get sick and others don´t?

Can you find healthy and sick crabs in the same area?

Response 9: Thank you for pointing this out. The progression of the HPND is relatively long, and the incidence of ponds is generally 30% to 40%, and it can reach 90% to 100% in severe cases. In addition, we collected biological samples from two adjacent ponds, which ensured that their growing environment and dietary conditions were consistent. We have added corresponding content to the article. Furthermore, we did not find some molecules in the control group that could support the possible HPND-related involvement of the pesticides.

Point 10: The paragraph of the 4.2 section is too long, please separate it for better reading.

Response 10: Thank you for pointing this out. We have segmented section 4.2 for better reading.

Point 11: Authors need to rephrase the conclusions in one general conclusion, instead of summarizing each analysis again. Authors claim that environmental stress, pesticides, and abiotic factors may be involved in HPND.

Response 11: Thank you for pointing this out. We have made appropriate revisions to the conclusion section of the article.

Point 12: Figure 1. Please correct units, it should be µm instead of um, and separate numbers from the units.

Response 12: Thank you for pointing this out. We have made the correct modifications in Figure 1.

Point 13: Figures 2, 3, 5, and 6. Figures are not readable, please modify them to increase quality. These figures could be saved as TIFF directly from R software.

Response 13: Thank you for pointing this out. We have edited these figures to improve the quality.

Point 14: Figure 3. Please add A) to the first figure and B) to the second in the figure legend. Also, it could be better to arrange figure 3 vertical format.

Response 14: Thank you for pointing this out. We have added A and B to the two pictures and arranged them vertically.

Point 15: Figure 6 could be replaced by a heatmap. Triangles and spheres are in black in the reference legend, however, the figure is green to red. Graphs could be arranged vertically for better presentation.

Response 15: Thank you for pointing this out. We regret that we still can't use a heatmap to replace Figure 6, but we have the pictures arranged vertically. We are sorry for the inadequacies in our work.

Point 16: Figure 7. Check the word in the figure, several typos.

Example “Molocular”

Response 16: Thank you for pointing this out. We have corrected the typos in the Figure 7.